



# GIS dataset: geomorphological record of terrestrial-terminating ice streams, southern sector of Baltic Ice Stream Complex, last Scandinavian Ice Sheet, Poland

Izabela Szuman[1], Jakub Z. Kalita[1], Marek W. Ewertowski[1], Chris D. Clark[2], Stephen J. Livingstone[2], Leszek Kasprzak[1]

[1]Faculty of Geographical and Geological Sciences, Adam Mickiewicz University, Poznań, 61-680, Poland
[2]Department of Geography, University of Sheffield, Sheffield, S3 7ND, UK

*Correspondence to*: Izabela Szuman (szuman@amu.edu.pl)

**Abstract.** Here we present a comprehensive dataset of glacial geomorphological features covering an area of 65 000 km$^2$ in central west Poland, located along the southern sector of the last Scandinavian Ice Sheet, within the limits of the Baltic Ice Stream Complex. The GIS dataset is based on mapping from a 0.4 m high-resolution Digital Elevation Model derived from airborne Light Detection and Ranging data. Ten landform types have been mapped: Mega-Scale Glacial Lineations, drumlins, marginal features (moraine chains, abrupt margins, edges of ice-contact fans), ribbed moraines, tunnel valleys, eskers, geometrical ridge networks and hill-hole pairs. The map comprises 5461 individual landforms or landform parts, which are available as vector layers in GeoPackage format at *http://doi.org/10.5281/zenodo.4570570* (Szuman et al., 2021a). These features constitute a valuable data source for reconstructing and modelling the last Scandinavian Ice Sheet extent and dynamics from the Middle Weichselian Scandinavian Ice Sheet advance, 50 – 30 ka BP, through the Last Glacial Maximum, 25 – 21 ka BP and Young Baltic Advances, 18 – 15 ka BP. The presented data are particularly useful for modellers, geomorphologists and glaciologists.

## 1 Introduction

Present-day ice sheet mass loss is a key driver of global average sea-level rise (Rignot et al., 2019). A key challenge is to quantify the sensitivity of ice sheets to climate change to improve future predictions of ice sheet mass loss and sea level rise. However, the observational time scale for contemporary ice sheets is short (decades) and it is a challenge to access their beds, contrary to the footprint of their palaeo-analogues (Stokes and Clark, 2001). Reconstructions of the configuration and evolution of palaeo-ice sheets through glacial cycles therefore provide a critical long-term constraint for testing and calibrating ice sheet models (e.g. Hughes, 2009;Stokes et al., 2015 and the references therein;Patton et al., 2016a;Patton et al., 2017;Ely et al., 2019a).

Geomorphological mapping is a powerful tool for reconstructing the glacial history of formerly glaciated areas (Clark, 1997;Kleman et al., 1997;Kjær et al., 2003), especially where *in situ* dating results (e.g., OSL, cosmogenic nuclides dating,



C14) are too sparse to reproduce highly dynamic ice sheet behaviour. Additionally, geomorphological mapping is widely used for shedding light on glacial processes, the evolution of landforms and for reconstructing ice sheets dynamics (e.g. Lukas, 2006;Kleman et al., 2008;Ó Cofaigh et al., 2008;Hughes, 2009;King et al., 2009;Kleman et al., 2010;Spagnolo et al., 2014;Ely et al., 2016;Evans et al., 2016;Jamieson et al., 2016;Chandler et al., 2018). A difficulty in acquiring accurate information on glacial landforms lies in the degradation of their original forms by postglacial processes, including water

erosion, aeolian activity, permafrost, or slope processes and human activity (Spagnolo et al., 2014). The results of ice sheet modelling are increasingly compared with or improved by incorporating geomorphological and geological field data (Stokes and Tarasov, 2010;Patton et al., 2017) and tools have been developed to aid such comparisons (e.g. Napieralski et al., 2006;Napieralski et al., 2007;Ely et al., 2019b).

The widespread presence of cross-cutting lineations has enabled reconstructions of the last Scandinavian Ice Sheet's (SIS)
growth and decay, relative switches in ice flow and the identification of ice streams (Kleman et al., 1997;Punkari, 1997;Boulton et al., 2001). The Baltic Ice Stream Complex (BISC) was one of the most prominent of the SIS ice streams. Typically, results from empirical and numerically modelled reconstructions (Kleman et al., 1997;Punkari, 1997;Houmark-Nielsen, 2010;Patton et al., 2016a;Patton et al., 2017) show the BISC as a vast area of highly dynamic, fast-flowing ice discharging towards the southern margin in northern Europe. However, there is lack of empirical data constraining the

southern margin of the BISC that can lead to ambiguities in understanding its behavior, which has resulted in millennial scale time differences between alternate reconstructions (cf. Hughes et al., 2016;Stroeven et al., 2016).

The specific role of ice streams in ice sheet mass balance is somewhat controversial. They are responsible for ~90% of the discharge of the Antarctic Ice Sheet (Bamber et al., 2000) with Pine Island and Thwaites ice streams contributing 32% of the discharge (Bamber and Dawson, 2020). Dynamic variations such as ice stream slowdown or stagnation can lead to a positive

mass balance (Joughin and Tulaczyk, 2002); for example, the Kamb Ice Stream stagnated about 150 years ago (Retzlaff and Bentley, 1993). Flow instabilities within ice sheets can lead to rapid ice discharges, with the Hudson Strait Ice Stream of the Laurentide Ice Sheet, for example, undergoing catastrophic purge events of accelerated ice discharge in response to positive basal thermal conditions, which led to increased meltwater production and thawing of soft subglacial sediments (MacAyeal, 1993;Rahmstorf, 2002). When considered across wider temporal and spatial scales ice streams can be thought of as drainage

networks, and for the Laurentide Ice Sheet these networks have been shown to scale fairly predictably with ice sheet volume (Stokes et al., 2016).

In this paper we use a new high-resolution digital elevation model to produce a comprehensive, high-quality geospatial dataset containing updated and newly discovered glacial landforms along the Polish southern sector of the SIS. These data can be used to reconstruct basal conditions, the pattern and style of retreat and ice flow dynamics during previous

glaciations. This area is of particular interest as it was at the southernmost limit of the BISC, which was highly active during the last glaciation and a key control governing SIS drainage and collapse (Patton et al., 2017).



## 2 Study area

The southern sector of the SIS was drained by the so-called BISC (Boulton et al., 2001, 2004;Kalm, 2012), which comprised several branches of fast flowing ice across the central European plains (see Punkari, 1997;Boulton et al., 2001;Stokes and Clark, 2001). The region of central west Poland was glaciated: (1) during the Middle Weichselian c. 30 - 50 ka BP (Wysota et al., 2009 and the references therein;Houmark-Nielsen, 2010;Hughes et al., 2016); (2) the Late Weichselian local Last Glacial Maximum (LGM), that took place between 25 and 21 ka BP (Tylmann et al., 2019); and (3) partly during two Young Baltic advances after 18 ka BP (Kjær et al., 2003;Stroeven et al., 2016). The southernmost position of the SIS margin, in central west Poland, was slightly south of 52ºN (Fig. 1) and was attained during the local LGM (Leszno/Brandenburg Phase). The Young Baltic advances are shown to be correlated with the Poznań/Frankfurt Phase and the Pomeranian Phase (Stroeven et al., 2016). The retreat of the ice sheet northwards across the Baltic Sea occurred at about 16.5 ka (Stroeven et al., 2016).

The study area covers c. 65 000 km$^2$ of central west Poland. This region was occupied by ice streams flowing from the north west (from Lower Odra region – LOR) and north east (Lower Wisła region – LWR) (Fig. 1). The earliest studies focused on glacial landform mapping across the study area were based on geomorphological field mapping by the Prussian Geological Survey at the end of the 19th and beginning of the 20th centuries (e.g. Berendt and Keilhack, 1894;Korn, 1912;Assmann and Dammer, 1916;Woldstedt, 1935). Further studies comprised geomorphological field mapping and analogue topographic maps analysis (Krygowski, 1947;Kozarski, 1959;Galon, 1961;Bartkowski, 1962;Kozarski, 1962;Bartkowski, 1963;Karczewski, 1963;Krygowski, 1963;Bartkowski, 1964, 1967, 1968;Stankowski, 1968;Bartkowski, 1969;Karczewski, 1971;Bartkowski, 1972;Kozarski, 1978;Karczewski et al., 1980;Liedtke, 1981;Rotnicki and Borówka, 1989, 1994;Kozarski, 1995). The first remote mapping attempts were based on Digital Elevation Models (DEM) with a resolution of 30 m and topographic maps with scales of 1:50000 and 1:10000 (Ewertowski and Rzeszewski, 2006;Przybylski, 2008). Streamlined bedforms along Stargard Drumlin Field, in the north western portion of the study area, along the LOR, were investigated by Keilhack (1897), Karczewski (1976) and more recently, by Hermanowski et al. (2019) (see Fig. 1B).



Figure 1: (a) An overview of Baltic Ice Stream Complex, southern sector of the SIS with B1-B4 ice streams after Punkari (1997). (b) Study area including the extent of previous studies that have mapped streamlined bedforms and major marginal spillways. Dashed lines indicate a bounding envelope for streamlined bedforms detected during previous studies. The difference between bounding envelopes for studies with overlapping areas indicate that a DEM pixel size of 30 m produces ambiguity in the detection of streamlined bedforms near the local LGM.



## 3 Methods

We used high-resolution LIDAR point cloud data (GUGiK, 2017) to generate a DEM with a 0.4 m ground sampling distance (GSD) across an area of about 250 x 300 km. Raw LIDAR data were distributed in square tiles of 1 x 1 km, with an average
survey point density between 4 and 6 pts/m2. Several highly urbanised complexes were covered by 0.5 x 0.5 km tiles with a density of 15-20 pts/m. The terrain coordinate system was EPSG:2180 (Table 1). The original dataset comprised about 65 000 raw data files and therefore required scripting to optimise the DEM generation process.

**Table 1: Details of EPSG:2180 coordinates system**

| Parameter | Value |
|---|---|
| EPSG code | 2180 |
| Reference ellipsoid | GRS80 |
| Projection type | Transverse Mercator |
| Prime meridian | 19oE (y0= 500 000 m) |
| Standard parallel | 0o (x0= -5 300 000,00 m) |
| Scale factor at prime meridian | 0.9993 |
| Unit | m |

The original data files were filtered leaving only points classified as a part of ground surface. Since the bottleneck for pre-
processing was disk storage transfer, each file was compressed using the LASzip algorithm (Isenburg, 2013) reducing the storage size of the files by 7-25%. Both operations were performed as a single step using the las2las procedure of LAStools library (Hug et al., 2004). The DEM was generated using the PDAL library (PDAL Contributors, 2018), assigning terrain height values based on Shepard's inverse distance weighting algorithm (Shepard, 1968), with an empirically adjusted radius of 4 m based on their density.

Further raster processing steps were performed using the GDAL library (Warmerdam, 2008). Merging (*gdal_merge*) resulted in more than 200 16x16 km tiles. Hillshade models were dynamically generated in Quantum GIS (*qgis.org*) using multidirectional sun azimuths from 0 to 90º and from 270 to 360º with increments of 45º; a sun altitude of 45º, and variable vertical exaggeration factor from 15 to 45 (estimated empirically for each subregion to emphasize small landforms). Such an approach aimed to improve landform recognition (Smith and Clark, 2005), enabling more accurate detection of landforms of
different orientation compared to investigations limited to only two orthogonal directions.

A second raster dataset was produced by de-trending the DEM data. Mean elevation was derived using a bilinear resampling algorithm across a 2 x 2 km window (*gdal_translate*) and this was subtracted from the original terrain elevation (*gdal_calc*). The de-trended images enabled glacial landforms on slopes to be better distinguished. Such an approach was not valid in the



vicinity of meltwater valleys where steep slopes resulted in underestimation of the mean value used in the subtraction and
thus disturbed the results.

The mapping process was based on semi-transparent hillshade images superimposed on the DEM and de-trended model. The DEM, de-trended and hillshaded raster files were enhanced with pyramids (*gdaladdo*) to facilitate map browsing over such a large area. Three operators independently mapped the study area to minimise the risk of landform misinterpretation. The resultant dataset is a minimalistic version, comprising only landforms recognized and verified by all three operators.
Additional validation of uncertain bedforms was performed using raw point clouds and ground-truthing.

We classified the glacial landforms into eight groups: (i) streamlined bedforms, comprising Mega-Scale Glacial Lineations (MSGLs) and drumlins; (ii) major and (iii) minor marginal features both including, moraine ridges, abrupt margins, edges of ice-contact fans, with major marginal landforms marking terminal positions of ice streams or the main recessional phases, and minor marginal landforms comprising sequences of moraine ridges, delicate isolated crests and abrupt margins
delineating subsequent positions of the receding ice streams; (iv) ribbed moraines, (v) tunnel valleys (vi) eskers; (vii) geometrical ridge networks and (viii) hill-hole pairs.

The coordinate system of the result dataset was based on the coordinate system of the original point cloud dataset (Table 1). The projection zone width of 10°, designed to comply with the territory of Poland, resulted in the scale factor at the prime meridian producing length distortions of up to 70 cm/km at the E border of the study area. In the worst-case scenario,
measurement of streamlined bedforms at the borders of the study area and oriented longitudinally resulted in an elongation error of 0.07%. Taking into consideration overall uncertainty of the dataset (see Section 5), such error is irrelevant for analyses focused on the distribution and morphological characteristics of glacial landforms (Clark et al., 2009;Spagnolo et al., 2014;Ely et al., 2016) or tying numerical modeling with their morphology (Jamieson et al., 2016).

## 4 Results of mapping – A GIS dataset of glacial features

In this study, we have produced a GIS dataset of glacial features, focused on landforms that provide information on ice dynamics. The dataset is available here: *http://doi.org/10.5281/zenodo.4570570* (Szuman et al., 2021a) and contains 5461 features in Geopackage format (*geopackage.org*). The glacial geomorphology map prepared from the dataset is presented in Fig. 2 and available via the Supplementary Data. A full morphological description and interpretation of the landforms is presented elsewhere (Szuman et al., 2021b); here, we outline the general characteristics of landforms included in the dataset
(Fig. 3, 4).



**Figure 2: Glacial geomorphology of the southern sector of the last Scandinavian Ice Sheet, central west Poland. This is designed to be viewed at a scale of 1:500000 and will need downloading from supplementary material (Fig. S1 at** *http://doi.org/10.5281/zenodo.4570570*) **to identify all features (see Szuman et al., 2021a).**

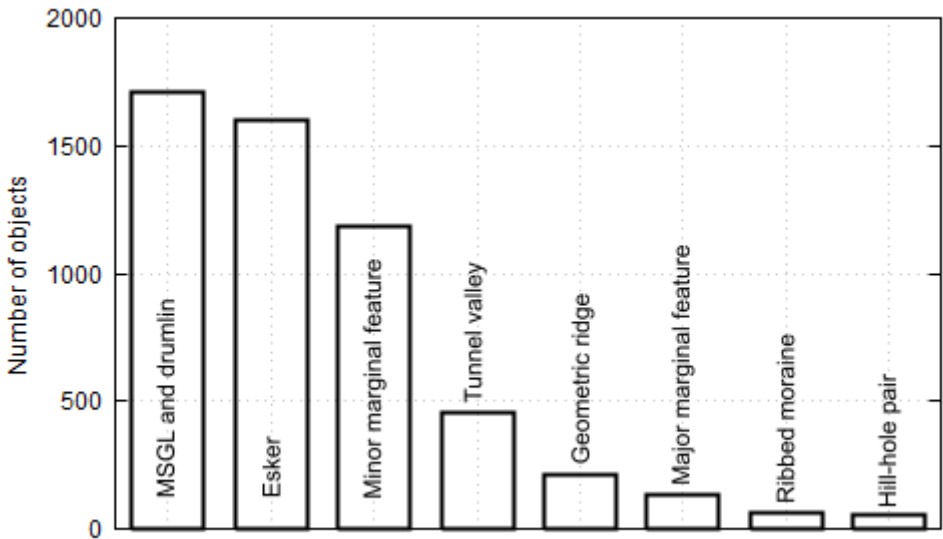


**Figure 3: Number of objects for each of the mapped landform groups.**

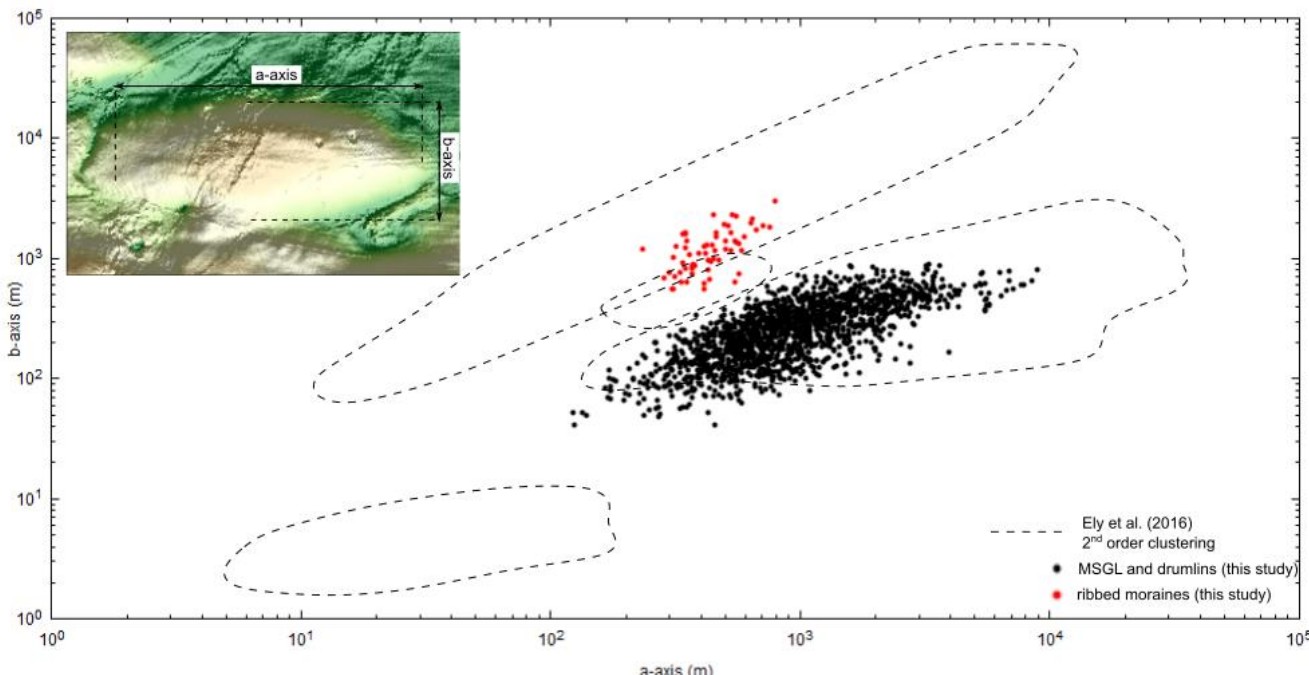

**Figure 4: Dimensions of mapped MSGL, drumlins and ribbed moraines on top of subglacial bedforms continuum presented by Ely et al. (2016) as an ellipsoidal geomorphological clusters.**





## 4.1 Streamlined bedforms: MSGLs and drumlins

MSGLs have a linear topographic expression that can be sustained for over 40 km in length (Clark, 1993), with typical lengths of between about 1 and 9 km, widths between 90 and 720 m, and spacing between 140 and 960 m (Spagnolo et al., 2014). MSGLs are one of the key geomorphological criteria for identifying former ice streams (Stokes and Clark, 2001), often occur with less elongated drumlins and have a convergent pattern of flow in the ice stream onset zone and fan-shaped divergence pattern near the margin (Stokes and Clark, 1999). A typical drumlin is a smooth, streamlined hill resembling the bowl of a spoon aligned longitudinally to ice movement direction (cf. Clark et al., 2009). Drumlin lengths are typically between 100 and 1000 m with widths typically three times smaller (Clark et al., 2009). Even though drumlins and MSGLs are often treated separately, taking their shape and size, they form a morphological continuum of subglacial lineations (Stokes et al., 2013;Ely et al., 2016) and therefore possibly share a similar origin (see also Stokes, 2018).

We mapped MSGLs and drumlins along their crests (Fig. 5, cf. Spagnolo et al., 2014;Ely et al., 2016). In the study area we identified 1712 MSGLs and drumlins with lengths of up to 10 km, widths between 50 and 800 m, and elongation ratios up to 1:60 (Fig. 3). They are mostly found in two areas: Stargard Drumlin Field in the NE of the study area, and the central part of the area between the Poznań and Leszno Phases (Fig. 1). Due to their association with high ice flow velocity (Jamieson et al., 2016) they were one of the key elements in the identification of 17 palaeo-ice streams (see Szuman et al., 2021b).

## 4.2 Major and minor marginal features

The marginal position of ice streams was identified from: (i) moraine ridges, (ii) abrupt margins, and (iii) edges of ice-contact fans. Moraine ridges are accumulations of glacial material formed at the ice margin during a recession, standstill or re-advance (e.g. Barr and Lovell, 2014). In our study, an abrupt margin relates to lateral or terminal margins of the former ice streams marked by distinct change in topography, i.e., a switch from streamlined terrain to ribbed moraine (e.g. Vérité et al., 2020;Szuman et al., 2021b), or a steep-sloped topographic step, formed at an ice contact-face or ice stream lateral margin (Patton et al., 2016b;Alley et al., 2019), without any distinctive moraine chains. An ice-contact fan is a cone of glacial and glacifluvial sediments formed at the ice margin (Benn and Evans, 2010). They are asymmetrical, comprising a gentle distal slope and steep proximal slope with a steep edge. Marginal landforms were classified as major or minor. Major marginal landforms are typically associated with the terminal moraines of ice streams, abrupt margins, main stillstands, or distinct lateral moraines (Fig. 6A). In the study area, the remnants of terminal moraines constitute either clear or vestigial arcuate ridges sporadically exceeding 10 m in amplitude and 2 km in width. Minor marginal landforms represent recessional and push moraines, and cupola hills with amplitudes up to several meters and widths typically below 400 m (Fig. 6B). Limited sedimentological investigations prohibited a more detailed genetic classification.





**Figure 5: Plain hillshade model (left) and DEM superimposed on hillshade model (right) with mapped (a) drumlins and (b) MSGLs as polylines along their crests. Where discontinuities in landforms (e.g., indicated with red X in Fig. 5B - hillshade) were interpreted as arising from postglacial activity from the action of water erosion, aeolian processes, or anthropogenic activity, the shape of the original landform was mapped.**

**Earth System Science Data Discussions**


**Figure 6: Examples of marginal features as a hillshade model (left) and DEM superimposed on hillshade model (right). (a) Cross-cutting of major marginal moraines. (b) Close-up on minor marginal moraines. Notice the prominent ice-marginal spillway disturbing the continuity of other glacial forms.**



### 4.3 Ribbed moraines

Ribbed moraines are parallel-spaced ridges, oriented transverse to the ice flow direction (Dunlop and Clark, 2006). They constitute sets formed subglacially in association with reduced ice flow velocities and typically found up-ice from drumlins and MSGLs (Ely et al., 2016), at ice streams shear margins (Vérité et al., 2020;Szuman et al., 2021b) or superimposed on streamlined bedforms (Stokes et al., 2008). In the study area, ribbed moraines form three elongated ribbons parallel to inferred ice flow direction located at the sides of areas interpreted as being occupied by ice streams (Szuman et al., 2021b).

The ribbed moraine ridges in the study area are up to about 400 m wide and 2 km long, with amplitudes of about 2 m (Fig. 7) and occupy areas up to 20 km long.

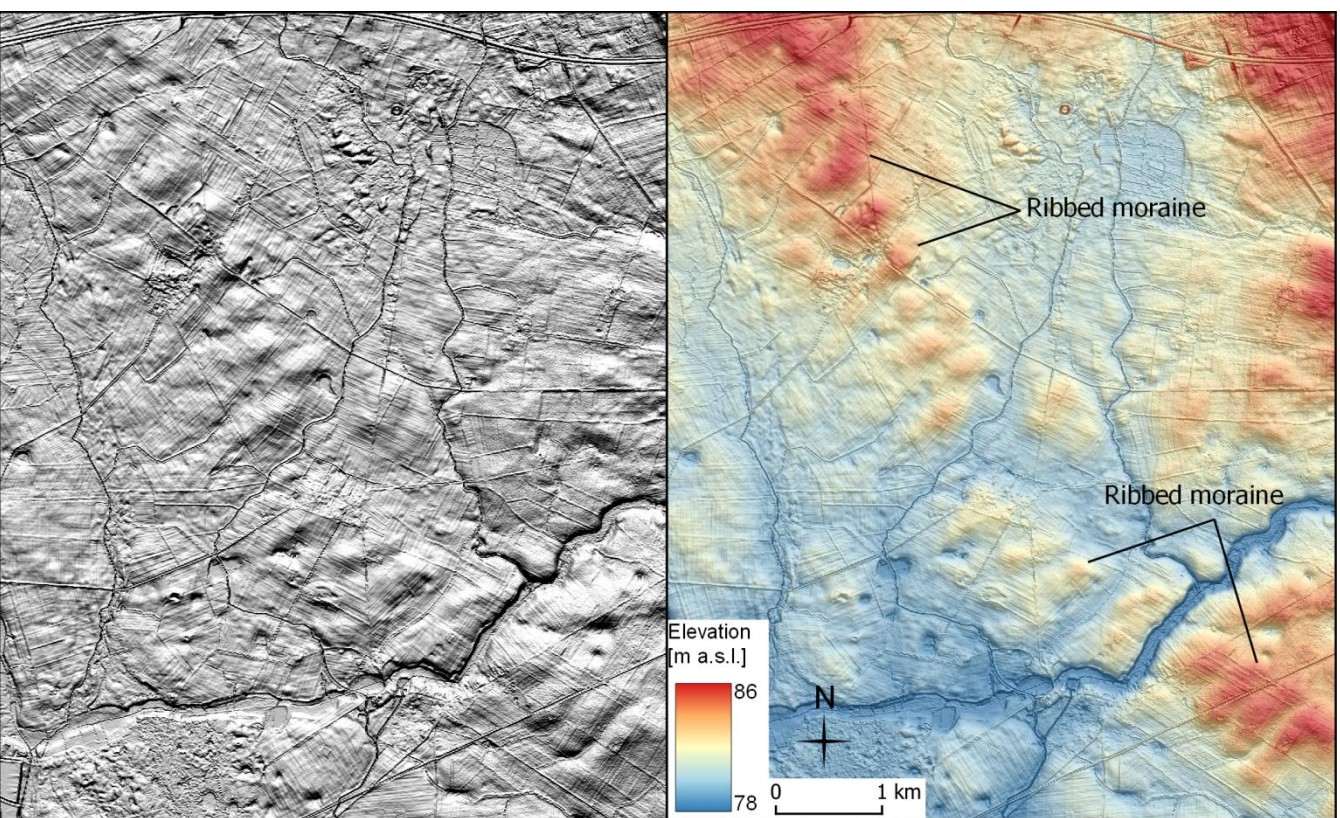

**Figure 7: Examples of ribbed moraine as plain hillshade model (left) and DEM superimposed on hillshade model (right).**

### 4.4 Hill-hole pairs

A hill-hole pair (Fig. 8A) comprises an ice-thrusted hill, concave up-glacier, located a short distance downstream from a depression of similar size and shape (Bluemle and Clayton, 1984;Benn and Evans, 2010). Hill-hole pairs are interpreted to form at the thin margin of advancing glaciers by plucking of large blocks of material from their frozen bed (Moran et al. 1980). Basal freezing induces compressive subglacial stresses leading to glaciotectonism (Moran et al., 1980;Rise et al., 2016). We identified about 50 of such features with relief of up to 50 m and typical widths between 1 and 4 km up to 15 km.





They are typically associated with ice stream margins and often occur near the terminal parts of tunnel valleys (see Livingstone and Clark, 2016).

## 4.5 Geometrical ridge network

The term geometrical ridge network is a non-genetic description of glacial landforms with an organised ridge pattern, either sub-parallel to each other (e.g. Rea and Evans, 2011) or cross-cutting at acute angles (e.g. Bennett et al., 1996), typically

inherited from ice structures. They are thought to originate from: (i) squeezing of subglacial sediments into basal crevasses (Evans and Rea, 1999;Evans et al., 2008;Evans et al., 2014); (ii) sediments melting-out from thrust planes and longitudinal foliation (Bennett et al., 1996;Glasser et al., 1998); or (iii) infilling of supraglacial crevasses with debris. For the features to be preserved, they require ice flow cessation and stagnation (e.g. Evans et al., 2016). Their preservation potential is low, so they are rare in glaciated areas comparing to other landforms. We identified 214 linear ridges with lengths up to 3 km,

amplitudes up to 10 m and spacing up to 1 km developed transverse to the ice flow direction. In the study area, they can be found superimposed on MSGLs (Fig. 8B).

## 4.6 Eskers

Eskers are sinusoidal ridges, usually orientated parallel to the ice flow, and composed of glacifluvial sands and gravels (Benn and Evans, 2010) deposited typically by meltwater in subglacial channels, and less often in englacial and supraglacial

channels (Hebrand and Åmark, 1989). They are often interpreted to form time-transgressively, close to a retreating margin (Kleman and Borgström, 1996;Storrar et al., 2014a, b;Livingstone et al., 2020). In our study, round-, sharp- and flat-crested eskers (see Perkins et al., 2016) are present. They constitute either single or multiple-crested forms, sporadically forming dendritic networks of ridges (e.g., Fig. 9). The eskers in the study area are relatively short, rarely exceeding lengths of 10 km in comparison to eskers described from the bed of the Laurentide Ice Sheet (Storrar et al., 2014b). They are especially

abundant in the northern part of the study area, related to the Pomeranian Phase (Fig. 2), but occur less frequently near the LGM margin.



**Figure 8: Examples of (a) hill-hole-pairs and (b) geometrical ridge network, as plain hillshade model (left) and DEM superimposed**
**on hillshade model (right).**

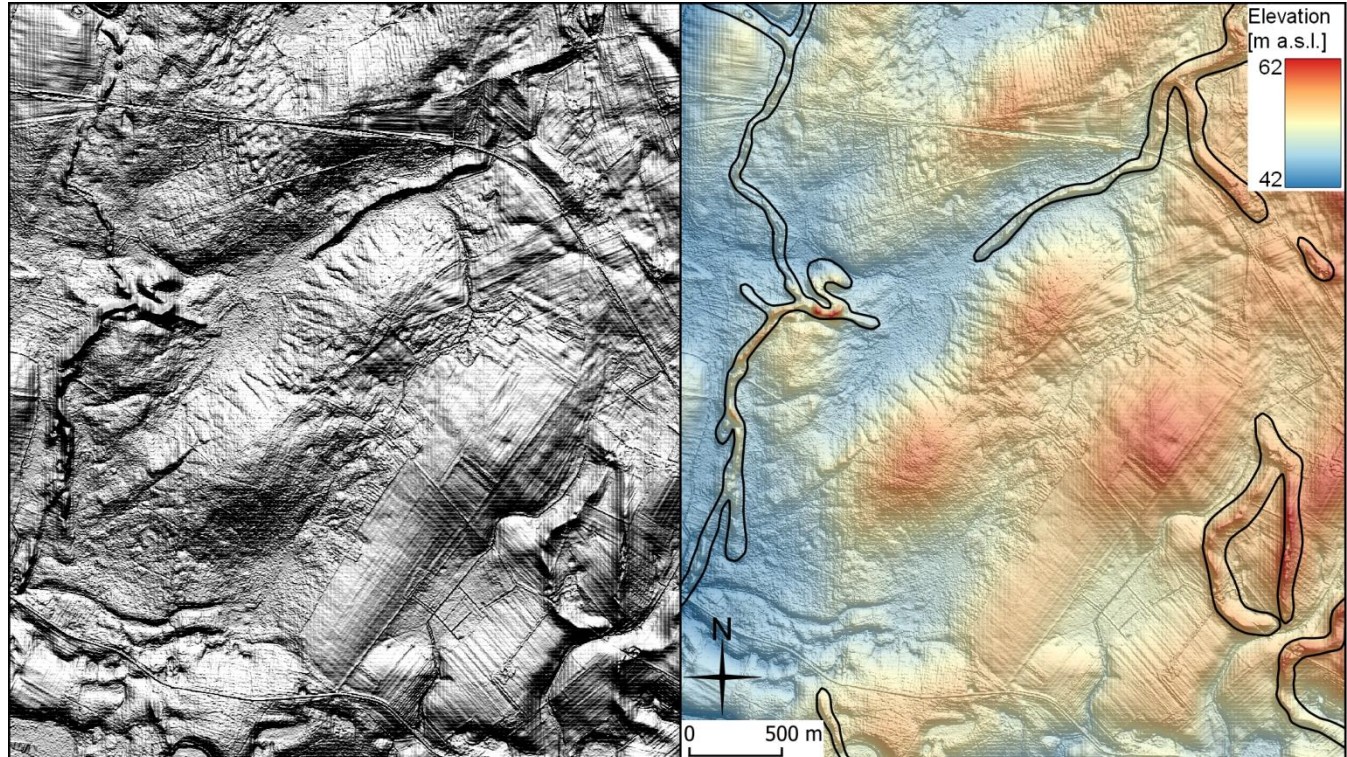

**Figure 9: Examples of eskers, along a drumlin field, as plain hillshade model (left) and DEM superimposed on hillshade model (right).**

### 4.7 Tunnel valleys

Tunnel valleys are formed subglacially due to meltwater erosion as a result of either outburst floods or by a gradual/steady state mechanism (Ó Cofaigh, 1996;Livingstone and Clark, 2016). The subglacial water flow is driven by gradients in hydraulic potential (function of the bed and ice elevation) (Shreve, 1972), resulting in orientations typically parallel to glacier flow (Kehew et al., 2012). Tunnel valleys are often associated with eskers, terminate near former ice margins and have undulating long profiles (Kehew et al., 2012). We identified 456 tunnel valleys in the study area (Fig. 10). The main

tunnel valleys have a quasi-regular spacing of about 5 and 10 km across the margins of the main glaciation phases (Figs 1, 2), with most of them occurring across the Poznań and Pomeranian margins. They often exceed 20 km in length, which is high (cf. Ottesen et al., 2020). However, their typical width below 2 km is conformant with counterparts of the Scandinavian Ice Sheet (cf. Jørgensen and Sandersen, 2006).



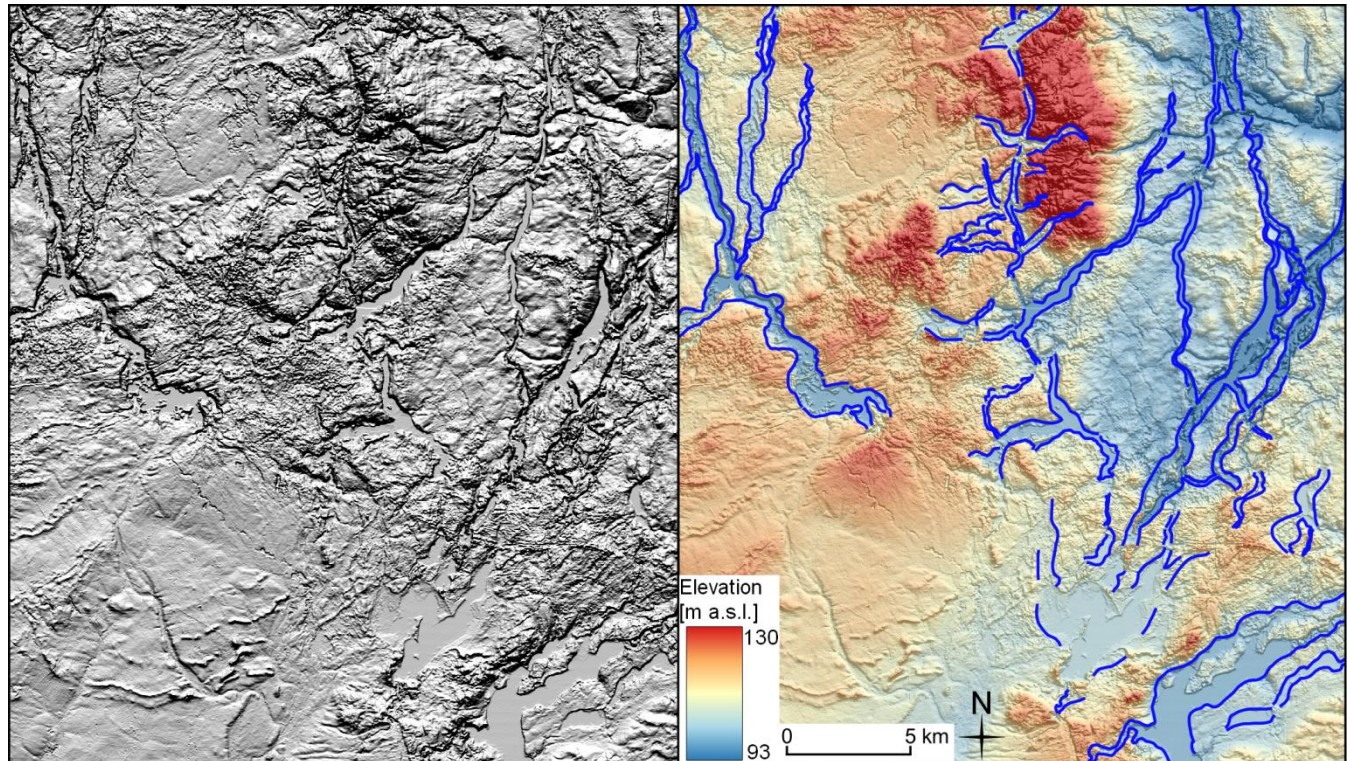

**Figure 10: Examples of tunnel valleys as plain hillshade model (left) and DEM superimposed on hillshade model (right).**

## 5 Limitations and uncertainty of the dataset

As noted by Spagnolo et al. (2014), increasing the scale of the DEM enabled more accurate recognition of low relief and fragmented forms. It also improved the identification of the bounding break of slopes of streamlined bedforms, enabling more accurate geomorphological parametrization (Clark et al., 2009). As shown by Napieralski and Nalepa (2010), decreasing the grid cell size from 10 to 1 m does not improve the identification of drumlins. However, it does improve the identification and geomorphological interpretation of more delicate forms, such as geometrical ridge networks, small eskers, minor moraines and narrow MSGLs (Figs 1, 12). As the resolution of the DEM model applied in this study (grid cell size of 0.4 m) far exceeds the dimensions of the indicated forms, no additional uncertainty due to model quality is introduced (cf. Hättestrand and Clark, 2006). The uncertainty of the resultant data lies in the degradation of the forms and misinterpretation during the mapping process. To minimize misinterpretations, we only added a feature to the final dataset if it was indicated by three independent operators.

Subglacial forms in some parts of the study area are degraded by large-scale, human activity (forestry, farming, urbanization; cf. Przybylski, 2008;Spagnolo et al., 2016), which obscures the glacial geomorphological record (Fig. 11A). The scale difference between the degrading effect and the original form is the main factor determining the degradation potential.


**Figure 11. (a) Highly urbanized area on top of MSGLs interrupted by postglacial valley. Notice that the scale difference of the imprint for both groups results in small degradation potential; (b) There is an impression of delicate streamlined bedforms, which are in fact of anthropogenic origin. Ground truthing showed that they are forestry artefacts.**

Delicate streamlined bedforms, such as some MSGLs with expected widths and spacing below 100 m reach the scale of
anthropogenic terrain modifications, potentially leading to single cases of misinterpretation. For example, field verification
showed that some landforms initially interpreted as MSGLs, were in fact old forestry roads (Fig. 11B). Many landforms are
partly or fully covered by aeolian sediments (dunes; Fig. 5A) that hampered landform interpretation. The continuity of many
of the mapped bedforms in our dataset is broken by water erosion (Fig. 5B). In particular, two prominent E-W oriented
spillways cross inferred paths of streamlined bedform fields (Figs 1, 6). The classification of ridges along large hummocky
areas was also problematic as delicate landforms have been partly buried (Fig. 2; cf. Hättestrand and Clark, 2006).

**Figure 12. (a) Examples of delicate MSLGs and geometric ridge networks on DTED2 30 m (left) and LiDAR-based 0.4 m DEM; (b) Examples of eskers near the LGM margin using DTED2 30 m (left), 1:10,000 contour-derived 5 m (middle) and LiDAR-based 0.4 m DEM. Notice how the higher-resolution model improves landform identification and interpretation.**

Since the interpretation of post-glacially degraded forms is challenging, they are often excluded from analyses (e.g. Spagnolo et al., 2014). In our dataset, the original shape of the forms was approximated and extrapolated where possible based on a comparison between terrain profiles and profiles expected for a particular form. Taking into consideration the general level of degradation such an approach increases the number of complete objects in the resultant dataset.

## 6 Significance of the data set

Compared to previous mapping efforts based on low-resolution data (30-100 m GSD), we use a 0.4 m Lidar DEM, that allowed for much more detailed mapping. This has resulted in the identification of completely new sets of landforms. We have mapped 5461 glacial landforms or landform parts in the southern sector of the last Scandinavian Ice Sheet. Despite some degradation of landforms, the study area represents one of the few regions in onshore Europe with well-preserved assemblages of MSGLs (Spagnolo et al., 2016). These data constitute a valuable source for reconstructing ice dynamics and

constraining modelling of the southern sector of the last Scandinavian Ice Sheet, which drained the large Baltic Ice Stream Complex and was rich in terrestrially-terminating ice streams (see Szuman et al., 2021b). In addition, the southernmost sector of the Scandinavian Ice Sheet responded sensitively to climate changes (Kozarski, 1962;Patton et al., 2016a;cf. Stroeven et al., 2016), and therefore provides an analogue for modelling the future response of the Greenland and Antarctic Ice Sheets. The produced dataset is useful for researchers working on ice stream dynamics, the response of ice to climatic

changes and subglacial landform formation. These data allow comparison between subglacial landforms between localities, including both formerly glaciated and modern glaciated areas occupied by ice streams.

## 7 Data availability

The data set can be accessed at *http://doi.org/10.5281/zenodo.4570570* (Szuman et al., 2021a).

## Authors contribution

IS, MWE, LK conceived the project. IS led the project and wrote the initial version of the manuscript, subsequently improved by the contribution of all co-authors. IS, JZK, MWE, mapped the landforms. All co-authors contributed to the interpretation of problematic areas.

## Funding and acknowledgements

This work was supported by the Polish National Science Centre (NCN) under Grant [2015/17/D/ST10/01975]. Chris Clark

and Stephen Livingstone were supported by the PalGlac project funded from the European Research Council (ERC) under the European Union's Horizon 2020 research and innovation programme (Grant agreement No. 787263)





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
