# Peer review of "GIS dataset: geomorphological record of terrestrial-terminating ice streams, southern sector of Baltic Ice Stream Complex, last Scandinavian Ice Sheet, Poland"

_Earth System Science Data, 2021_

## Author Comment (AC2)

Answer to Referee 1

Dear Referee,

We appreciate your overall positive feedback on the manuscript and valuable comments. We have corrected the manuscript according to your suggestions and provide answers for specific comments below.

General comments

The manuscript provides details on a new GIS dataset of geomorphological mapping relating to the Baltic Ice Stream Complex of last Scandinavian Ice Sheet. This geomorphological map adds significantly to previous mapping in the region covering a larger area for a more holistic perspective on ice stream/lobe interactions, and using a higher resolution DEM to add further details to previously mapped areas. Publication of this GIS dataset will allow further interrogation of the complex set of glacial landforms in the region and enable assimilation with geomorphological mapping of other areas of the Scandinavian Ice Sheet. The dataset is therefore of value for future use. The dataset is easily accessible, of high quality and is complete. Sources of error/mis-interpretation are discussed in the manuscript. Overall, I find the manuscript well-written and supports the GIS dataset. I have very few comments to add and those below are all relatively minor points.

Specific comments

1. Introduction – I realise that section 6 is a 'significance' section, but I think you could add a sentence or two in the introduction to say why publishing the GIS dataset is important or what you hope this will help achieve/facilitate. This could perhaps also be emphasised a little more in the significance section.

*Added the following sentences to introduction:*

*"It can also be easily integrated with data from other areas of the SIS to improve palaeo-glaciological reconstructions."*

*and*

*"These data can be used as a valuable source of information on the behaviour of terrestrially-terminating ice streams that have no contemporary analogues."*

2. Lines 44-46: It would be helpful to expand a little on how there are such large time differences if there is a lack of any data – presumably there is some data somewhere to result in these differences? You could also perhaps add a sentence as to how this new mapping could help to resolve these issues/improve understanding of BISC behaviour.

*We have changed the cause of ambiguity from lack of data to sparse geochronological and geomorphic data. The sentence is now as following:*

*"However, sparse geochronological and geomorphic data constraining the southern regional margin reconstructions of the BISC, can lead to ambiguities in understanding its behaviour, which has resulted in millennial scale time differences between alternate ice sheet reconstructions (cf. Hughes et al., 2016;Stroeven et al., 2016)."*

3. Line 119: Three operators independently mapped the study area – could you say more about how this worked – was there one operator who consistently identified more and one who consistently identified less? – if so I guess the mapping is really based on one more cautious operator. Or was it

variable as to which operator did or did not recognise a feature? Was there any discussion about a landform if two operators had both mapped it but one hadn't? What I'm getting at is could genuine landforms have been missed (and to what extent) as a result of this rigorous process to avoid false positives?

*Three people mapped the area separately on three computers. One of them was merging the data into one dataset. The differences were discussed and features removed if no agreement was reached (the number of such cases was low considering the dataset size) or included in case of consensus. For some uncertain areas raw point cloud data was additionally analysed. The area that we discussed a lot is indicated in Figure 11. The potential interpretations were that the elongated landforms are either glacial or processing based error during DEM generation as the area is mostly covered with forests. Ground truthing showed that they are anthropogenic forestry artefacts.*

*Added the following sentence into methods section:*

*"The differences were discussed and features removed if no agreement was reached (the number of such cases was marginal considering the dataset size) or included in case of consensus."*

4. Section 4.2 I have made a few comments below in the technical corrections relating to moraine formation and terminology. This section just needs a few minor adjustments in relation to this. As far as I can see it doesn't affect the GIS dataset.

Technical corrections

Line 31: 'ice sheet dynamics'

*Corrected.*

Line 44: southern margin of what? The SIS? The Baltic Sea?

*Added "… margin of SIS…"*

Line 50 is the Kamb Ice Stream example relevant to this sentence? – is there evidence its stagnation caused a positive mass balance?

*In order to clarify, we changed the sentence to:*

*"Dynamic variations such as ice stream slowdown or stagnation can lead to a positive mass balance; for example, slowdown of Whillans Ice Stream caused a switch to a positive mass balance of Siple Coast ice streams (Joughin and Tulaczyk, 2002)."*

Line 123: Add 'ice' marginal features

*Added.*

Line 167: This should be 'active recession'. Technically, moraines don't form during recession – they only form if there is a minor advance or stillstand during overall recession.

*Thank you. Changed the fragment to: "at the overall receding ice margin during standstill or re-advance"*

Line 174: what is an abrupt margin defined as? Also, how do you define a stillstand or differentiate this from a terminal moraine? – is this based on available dates?

*As noted in Section 4.2, an abrupt margin is defined as a "distinct change in topography, i.e., a switch from streamlined terrain to ribbed moraine (e.g. Vérité et al., 2020;Szuman et al., 2021b), or a steep-*

*sloped topographic step, formed at an ice contact-face or ice stream lateral margin (Patton et al., 2016b;Alley et al., 2019), without any distinctive moraine chains."*

*Since the scope of the study did not concern sedimentological investigations, the dataset does not contain detailed genetic classification. We instead define morphologically as: "major marginal feature" and "minor marginal feature" in it. However, our aim was to emphasise the presence of ice streams and the character of the terminal moraines that can be inferred by their lobate geometry and association with lineation flowsets.*

*We have changed the sentence by adding association to lineations part:*
*"In the study area, the remnants of terminal moraines constitute either clear or vestigial arcuate ridges sporadically exceeding 10 m in amplitude and 2 km in width and are sometimes associated with lineation flowsets."*

Line 176: I find the phrase 'recessional and push moraines' a bit strange because recessional moraines could be push moraines (or they might not be) – they are not one or the other. Recessional means that they form during overall recession, whereas push moraines relate to a genetic formation. Is there any sedimentological evidence that they are push moraines? – if not, I would be tempted to just call them recessional (this encompasses any minor advances that could form a push moraine).

*Changed to recessional moraines.*

Line 251: 'interpretation of [smaller land]forms' (also Line 264 – is 'delicate' the right word? – from a mountain glaciation perspective MSGL are pretty big).

*Changed to "smaller landforms".*

7 Data availability – Is it worth saying that the dataset is best viewed in QGIS, but can be opened and manipulated within ArcGIS (or words to that effect)?

*Added the following sentence to Data availability section:*
*"The dataset can be visualised and manipulated in any Geographic Information System software package capable of handling common spatial vector data formats."*

Figure 1: Any idea where the pre-local LGM ice sheet reached? Can either the LOR and LWR be added to Figure 1 and/or flow lobes B2 and B3 be added to the text after the LOR and LWR are introduced to better connect what is in the text to the study area map. Relating to the text in the figure caption - it looks like there are some streamlined bedforms outside of the dashed boxes? – e.g within the Przybylski (2008) box. I'm also struggling to follow what is meant by the final sentence of the figure caption – I assume that it means that studies of the same area have produced slightly different maps as a result of image resolution?

*We think that adding the pre-local LGM limit at this stage of understanding the behaviour of the SIS southern sector would be too speculative.*

*We now make the association to B2 and B3 ice streams while introducing the LOR and LWR in the text as well as LOR and LWR to Figure 1.*

*The streamlined bedforms in Figure 1b are those mapped in this study. Since the areas of previous studies overlap, placing the results of their mapping as box envelopes on top of our mapping results shows the progress enabled by using high resolution model for the mapping. At the same time while comparing overlapping study areas with the same model resolution – the subjectivity of the mapping process that is amplified by poor resolution can be roughly accessed.*

*In order to clarify, we have changed part of the figure caption to:*
*"b) Study area including the extent of previous studies that have mapped streamlined bedforms and major marginal spillways (boxes). Streamlined bedforms are from this study."*

Figure 8 caption- one hill hole pair. Fig 8b the geometrical ridge network doesn't look like it's superimposed on an MSGL here? The high elevation ridge looks more like a terminal moraine or esker? Perhaps I'm just viewing it out of context.

*Corrected the hill-hole caption according to the suggestion.*

*Thank you for that comment. The mistake was due to changes between versions of the figure and not correcting the associated text. We interpret the elevation ridge that is superimposed on the geometrical ridge network as an esker in the published dataset.*

*We have changed the sentence to:*

*"In the study area, they can be found in superimposition with MSGLs or eskers (Fig. 8B for the latter)."*

Answer to Referee 2

Dear Referee,

We appreciate your overall positive feedback on the manuscript and valuable comments. We have corrected the manuscript according to your suggestions and provide answers for specific comments below.

Review of "GIS dataset: geomorphological record of terrestrial-terminating ice streams, southern sector of Baltic Ice Stream Complex, last Scandinavian Ice Sheet, Poland" by Izabela Szuman et al.

This manuscript details glacial and geomorphological features for western Poland based on high resolution (0.4 m) DEM datasets. This work is critical for reconstructing the glacial history of the southern portion of the Scandinavian Ice Sheet and is therefore a valuable scientific contribution. My knowledge pertains largely to the documentation of the geomorphological record and its place within the greater context of glaciation in western Poland. I am unable to provide a thorough critique the specific methodologies employed in this study (e.g. extracting DEMs from high-resolution LIDAR point cloud data), however these methods appear to be sound and follow previously cited studies.

Overall, the manuscript is focused on methodology, documentation, and does an excellent job of providing examples of how each landform was calculated/measured, showing "comparison figures" between the plain hillshade model and resulting glacial geomorphological record that is identified in the database. The manuscript is well written, highlighting the major research gaps and the overall structure of the accompanying dataset.

Minor comments

1. L24- "a challenge to access their beds" – recommend adding more context here. To get a first-hand account of the ice-sheet and geomorphological processes?

*Added the following sentence to introduction:*
*"Geomorphological studies on glacial landforms provide opportunity to elucidate processes at the ice-bed interface responsible for their formation and that are crucial for understanding and modelling ice sheets (e.g. Stokes 2018)."*

2. L37 –" tools have been developed to aid such comparisons" – perhaps a brief mention of what types of tools?

*Changed the sentence to:*

*"The results of ice sheet modelling are increasingly compared with or improved by incorporating geomorphological and geological field data (Stokes and Tarasov, 2010;Patton et al., 2017) and tools that quantify the level of agreement between empirical data and modelling results have been developed to aid such comparisons (e.g. Napieralski et al., 2006;Napieralski et al., 2007;Ely et al., 2019b)."*

3. L48 "They are responsible for ~90% of the discharge of the Antarctic Ice Sheet (Bamber et al., 2000)" – Is it possible/appropriate to add a newer reference here given the amount of work that has taken place on ice streams in Antarctica since 2000 (e.g authors Rignot, Stokes, Livingstone…etc? ). Are there any newer studies or statistics that can be cited here?

*We have updated the references concerning the sentence with:*
*Rignot, E., Mouginot, J., Scheuchl, B., van den Broeke, M., van Wessem, M. J. & Morlighem, M. 2019: Four decades of Antarctic Ice Sheet mass balance from 1979–2017. Proceedings of the National Academy of Sciences 116, 1095.*
*Gardner, A. S., Moholdt, G., Scambos, T., Fahnstock, M., Ligtenberg, S., van den Broeke, M. & Nilsson, J. 2018: Increased West Antarctic and unchanged East Antarctic ice discharge over the last 7 years. The Cryosphere 12, 521-547.*

*The relevant sentence is now as follows:*
*"The specific role of ice streams in ice sheet mass balance is somewhat controversial. They are responsible for ~90% of the discharge of the Antarctic Ice Sheet (Bamber et al. 2000) with Pine Island and Thwaites ice streams contributing 32% of the discharge (Bamber & Dawson 2020)."*

4. Fig 1 –Perhaps outline the "this study" box in a thicker black line? It took a few reads to understand the outline of the study area compared to the previous studies.

*We have added thicker line for "this study" box.*

5. Fig 2 – Is it possible to draw the "study area" box in this figure? Also, while it is clear from the figure caption that this map is intended to be viewed in a higher-resolution, this this figure would benefit from a legend that explains the correspondence between lines and landforms.

*Thank you for this comment. We are aware that the quality of the figure in the manuscript possibly can confuse the reader as elements such as the legend present in the upper right part of the figure may not be that conspicuous. The extent of the figure corresponds to the study area. Therefore there is no need to add the "study area" box. However, we have added a caption to Figure 1 linking the study area box in Figure 1 to Figure 2.*